# A Scoping Review Evaluating the Current State of Gut Microbiota Research in Africa

**DOI:** 10.3390/microorganisms11082118

**Published:** 2023-08-20

**Authors:** Sara M. Pheeha, Jacques L. Tamuzi, Bettina Chale-Matsau, Samuel Manda, Peter S. Nyasulu

**Affiliations:** 1Division of Epidemiology and Biostatistics, Faculty of Medicine and Health Sciences, Stellenbosch University, Cape Town 7500, South Africa; pheeha.sm@gmail.com (S.M.P.);; 2Department of Chemical Pathology, Faculty of Medicine and Health Sciences, Sefako Makgatho Health Sciences University, Pretoria 0208, South Africa; 3National Health Laboratory Service, Dr George Mukhari Academic Hospital, Pretoria 0208, South Africa; 4Department of Chemical Pathology, Faculty of Health Sciences, University of Pretoria, Pretoria 0028, South Africa; 5National Health Laboratory Service, Steve Biko Academic Hospital, Pretoria 0002, South Africa; 6Department of Statistics, Faculty of Natural and Agricultural Sciences, University of Pretoria, Pretoria 0028, South Africa; 7Division of Epidemiology and Biostatistics, School of Public Health, Faculty of Health Sciences, University of the Witwatersrand, Johannesburg 2193, South Africa

**Keywords:** gut microbiota, gut microbiome, human health, diseases, dysbiosis, eubiosis, F/B ratio, gut diversity, gut richness, taxonomic profiles, Africa

## Abstract

The gut microbiota has emerged as a key human health and disease determinant. However, there is a significant knowledge gap regarding the composition, diversity, and function of the gut microbiota, specifically in the African population. This scoping review aims to examine the existing literature on gut microbiota research conducted in Africa, providing an overview of the current knowledge and identifying research gaps. A comprehensive search strategy was employed to identify relevant studies. Databases including MEDLINE (PubMed), African Index Medicus (AIM), CINAHL (EBSCOhost), Science Citation index (Web of Science), Embase (Ovid), Scopus (Elsevier), WHO International Clinical Trials Registry Platform (ICTRP), and Google Scholar were searched for relevant articles. Studies investigating the gut microbiota in African populations of all age groups were included. The initial screening included a total of 2136 articles, of which 154 were included in this scoping review. The current scoping review revealed a limited number of studies investigating diseases of public health significance in relation to the gut microbiota. Among these studies, HIV (14.3%), colorectal cancer (5.2%), and diabetes mellitus (3.9%) received the most attention. The top five countries that contributed to gut microbiota research were South Africa (16.2%), Malawi (10.4%), Egypt (9.7%), Kenya (7.1%), and Nigeria (6.5%). The high number (*n* = 66) of studies that did not study any specific disease in relation to the gut microbiota remains a gap that needs to be filled. This scoping review brings attention to the prevalent utilization of observational study types (38.3%) in the studies analysed and emphasizes the importance of conducting more experimental studies. Furthermore, the findings reflect the need for more disease-focused, comprehensive, and population-specific gut microbiota studies across diverse African regions and ethnic groups to better understand the factors shaping gut microbiota composition and its implications for health and disease. Such knowledge has the potential to inform targeted interventions and personalized approaches for improving health outcomes in African populations.

## 1. Introduction

The gut microbiota, a collection of microorganisms that inhabit the human gastrointestinal system, has emerged as a critical area of research in recent years [1]. The gut microbiota plays a crucial role in maintaining human health [2] by metabolizing proteins and complex carbohydrates [3], producing certain hormones and neurotransmitters [4], and synthesizing specific vitamins and essential amino acids, short-chain fatty acids [5,6], and certain lipopolysaccharides [7]. Moreover, the host’s innate and adaptive immune systems collaborate with the gut bacteria to preserve intestinal homeostasis [8] and reduce inflammation [9].

An imbalance in the gut microbiota (dysbiosis) has been linked to a wide range of diseases, including diabetes mellitus (DM) [10], obesity, and inflammatory bowel disease (IBD) [11]. In addition, the gut microbiota has been shown to have a significant impact on the immune system [12] and neurological function [13], making it a key area of research for scientists across a range of disciplines. The importance of gut microbiota research lies in its potential to improve human health outcomes [14]. By better understanding the composition and function of the gut microbiota, researchers can identify ways to prevent and treat diseases as well as optimize overall health and wellbeing. 

Motivated by the desire to understand the gut microbiota and its significance in human health, numerous scientists worldwide have extensively researched this complex subject. Therefore, the gut microbiota has been shown to vary across different populations and geographic regions [15], highlighting the importance of studying its diversity in different contexts. Such studies can help to identify population-specific risk factors and develop targeted interventions that are tailored to the unique needs of different populations. Despite the importance of gut microbiota research being highlighted by various authors [16], it is crucial to note that most of the research was conducted in Western populations, and therefore little is known about the gut microbiota of African populations [1,17]. Given the significant variations in dietary patterns, lifestyle practices, and the environment between African and Western populations, it is essential to explore the gut microbiota of African populations to develop a better understanding of the role of the gut microbiota in maintaining health in these populations. 

Moreover, the African continent is characterized by a range of socio-economic and demographic factors that influence gut microbiota composition. These factors include urbanization, Westernized diets [18], variations in hygiene practices [19], and genetic diversity among African ethnic groups [1]. Africa is also burdened with numerous health challenges, including increasing rates of non-communicable diseases [20] such as diabetes, hypertension, and obesity, as well as exposure to infectious diseases like malaria, human immunodeficiency virus/acquired immunodeficiency syndrome (HIV/AIDS), and tuberculosis [21]. The gut microbiota is emerging as a potential contributor to the aetiology and progression of these health conditions.

This scoping review aimed to evaluate the current state of gut microbiota research in Africa, focusing on the range of studies conducted across various African populations and settings. The evaluation included but was not limited to the identification of diseases studied in relation to the gut microbiota, the determination of methods used to profile the gut microbiota and study designs employed, as well as documenting generalized key findings of the included studies, therefore enabling us to associate the gut microbiota with health outcomes in African populations. By conducting this scoping review, we hoped to identify research gaps and areas for future exploration in this budding field of research. 

## 2. Methods

The current scoping review along with the protocol were conducted according to the JBI methodology for scoping reviews [22] and the Preferred Reporting Items for Systematic Reviews and Meta-Analysis extension for Scoping Reviews (PRISMA-ScR) [23]. The protocol for the scoping review has been previously published and can be accessed using the following link: https://www.mdpi.com/2409-9279/6/1/2 (accessed on 20 June 2023).

### 2.1. Scoping Review Questions

Among other inquiries, this scoping review primarily answered the questions outlined in the scoping review protocol [24].

### 2.2. Eligibility Criteria

Criteria for inclusion: The scoping review included studies conducted in African populations, without restrictions based on race, gender, age, health status, disease type, study type/design, or setting. The review considered studies examining various conditions if they characterized and described the gut microbiota. Articles that investigated the gut microbiota in apparently healthy individuals were also included. Additionally, studies that utilized any type of technology to profile gut microbiota from human samples were incorporated. The review encompassed experimental study designs and/or clinical trials. Furthermore, analytical observational studies, including prospective and retrospective cohort studies, case–control studies, and analytical cross-sectional studies, were included. Research articles that were exclusively published in English were incorporated, as our aim was to conserve human and financial assets. The task of locating appropriate individuals for translating articles from other languages into English is notably challenging and costly.

Criteria for exclusion: Studies that did not involve human participants or included both humans and animals were excluded from the review. Studies involving non-African populations, except for those utilizing existing/previously published data for comparison purposes, were also excluded. Studies that involved a mixture of African participants and participants from non-African countries, as well as those that did not clearly state or describe the country or population in which the study was conducted, were excluded. Additionally, studies that did not describe and characterize the gut microbiota were excluded. The review did not include any type of reviews, editorials, opinion papers, commentaries, news articles, or notes. 

### 2.3. Search Strategy

#### 2.3.1. Identifying Search Terms and Sources of Information

The search strategy aimed to locate both published and unpublished studies (i.e., conference proceedings/abstracts/posters). An initial limited search of PubMed was undertaken by our faculty librarian to identify articles on the topic. The text words contained in the titles and abstracts of relevant articles and the MeSH terms were used to develop a full search strategy. This was then adapted for each included database or information source (see Appendix A).

#### 2.3.2. Conducting the Searches 

An information specialist searched the following databases using the search strategy shown in Appendix A: MEDLINE (PubMed), African Index Medicus (AIM), CINAHL (EBSCOhost), Science Citation index (Web of Science), Embase (Ovid), Scopus (Elsevier), WHO International Clinical Trials Registry Platform (ICTRP), and Google Scholar.

We also searched sources of unpublished studies/grey literature, including conference proceedings. The reference lists of some included sources of evidence were also screened to identify additional studies.

### 2.4. Selection of Sources of Evidence

After the search process, all citations found were gathered and uploaded into EndNote, where the information specialist removed any duplicates. Subsequently, a compressed EndNote database and an RIS file used to import the references into Covidence were sent to one of the reviewers (S.M.P). Two independent reviewers (J.L.T. and S.M.P) screened the titles and abstracts of the identified articles against the eligibility criteria for the scoping review. This was followed by full-text screening of the studies independently conducted by S.M.P and J.L.T. In the case of any disagreements throughout the process, a third reviewer was consulted for resolution. Studies that did not meet the inclusion criteria were documented. A comprehensive presentation of the search results and the study inclusion process can be found in the PRISMA flow diagram [25] (Figure 1).

### 2.5. Data Charting and Data Items

The data from all the full-text articles included in the review were obtained by S.M.P using a specialized charting tool designed for this specific scoping review. The accuracy of the extracted data was verified by J.L.T. The extracted data consisted of various elements, such as the title of the study, the name of the first author and year of publication, whether the gut microbiota was described or not, the specific disease under study and its relevance to public health, details about the population involved, the country where the study took place, the study setting (urban or rural), the technologies employed to profile the gut microbiota, the study type/design, sample size, type of sample used, demographic characteristics including age range, gender ratio (female/male), and ethnicity/tribe, the aims and objectives of the study, interventions, funding sources, and the main/generalized findings of the studies (see Appendix A). Any disagreements that arose between the reviewers (J.L.T. and S.M.P) were resolved through discussion or with the assistance of additional reviewers. 

## 3. Results

Frequency statistics and graphs were generated using Stata 18 software (StataCorp. 2023. *Stata Statistical Software: Release 18*. College Station, TX: StataCorp LLC), and the map (Figure 4) was created with Microsoft Power BI Desktop.

### 3.1. Study Selection Process

A total of 3107 articles were initially identified through the database search. After removing duplicates, 2113 unique articles remained. The titles and abstracts of these articles as well as the records identified from citation searching and other sources (*n* = 23) were screened for relevance, resulting in the exclusion of 1950 articles. The full texts of the remaining 186 articles were assessed for eligibility, and ultimately 154 studies met the inclusion criteria and were included in the scoping review (Figure 1).

### 3.2. Description of the Generalized Population from All the Included Studies

The most studied population group were children (37.1%), which was followed by adults (27.9%) and then infants (10.4%). Most of the studies (90.3%) did not mention the ethnic groups/tribes in which the research was conducted. Of the ethnicities that were mentioned, blacks (2.6%) and mixed-ancestry individuals (1.3%) were mostly studied as compared to the rest of the groups. Most of the studies were dominated by females, as shown by the high number (*n* = 52) of ratios (female/male) between 1.05 and 5.83. However, 48 of the studies that were included in the current scoping review did not mention the number of males and females looked at. The age of the generalized population ranged from 0 months to 84 years (Table 1).

### 3.3. Characteristics of Included Studies

The included studies were published between the years 2010 and 2023. They encompassed 142 scholarly articles and 12 poster/abstract publications. The sample sizes varied across studies, ranging from 3 to 1900 participants (Appendix A). 

#### 3.3.1. Study Settings

Half of the studies included (50%) did not indicate the type of study setting in which their research was conducted. When looking at those that indicated the study setting, most of them were conducted in rural areas (27.9%) (Figure 2). 

#### 3.3.2. Disease Types

Of all the studies that were included in the current scoping review, only 11 diseases of public health significance (which were defined as the top causes of mortality in the African region as listed by the WHO in 2018 [26]) were studied in relation to the gut microbiota. The diseases included Alzheimer’s disease, breast cancer, coronavirus disease (COVID-19), chronic kidney disease, colorectal cancer, diabetes mellitus, diarrhoeal diseases, hepatitis C virus, HIV, malaria, and tuberculosis. The top three diseases that were mostly studied included HIV (14.3%), colorectal cancer (5.2%), and diabetes mellitus (3.9%), all of which formed part of the list of top causes of mortality in Africa. HIV was predominantly studied by South Africa (40.9%). Other countries that studied the infectious disease included Uganda (13.6%), Cameroon (9.1%), Mozambique (9.1%), Zimbabwe (9.1%), Ethiopia (4.5%), Ghana (4.5%), Kenya (4.5%), and Nigeria (4.5%). Colorectal cancer was mostly studied in Egypt (25%). Ethiopia (12.5%), Kenya (12.5%), Morocco (12.5%), Nigeria (12.5%), and Zimbabwe (12.5%) equally studied the condition as well, while the remaining 12.5% was accounted for by the “other” category. Fifty percent (50%) of the diabetes mellitus studies came from Egypt. Other countries that also studied the disease were Nigeria (16.7%), Sudan (16.7%), and Tunisia (16.7%). Most of the studies (42.9%) that were included in this review did not look at any disease in relation to the gut microbiota (Figure 3).

#### 3.3.3. Countries Where the Included Studies Were Conducted

The top five countries that contributed to gut microbiota research include South Africa (16.2%), Malawi (10.4%), Egypt (9.7%), Kenya (7.1%), and Nigeria (6.5%) (Figure 4).

#### 3.3.4. Methods/Techniques Used to Profile Gut Microbiota

The dominant method of choice for gut microbiota profiling was 16S rRNA/metagenomic sequencing (76.0%), which was followed by quantitative PCR/real-time PCR (9.7%) (Figure 5). 

#### 3.3.5. Study Types/Designs

The top five study types/designs which were employed by the included studies were observational/cross-sectional studies (38.3%), experimental studies/clinical trials (16.9%), case–control studies (9.1%), sub-studies (7.8%), and comparative studies (5.8%). Of the sub-studies identified, 75%, 16%, and 8.3% were nested within clinical trials, cross-sectional studies, and cohort studies, respectively (Figure 6). 

#### 3.3.6. Types of Samples

Most of the included studies used stool samples for gut microbiota profiling (90.3%), while the rest made use of rectal swabs/sponges (6.5%) and colonic mucosal samples/tissues (3.3%) (Figure 7).

### 3.4. Generalized Study Findings

Generally, some studies indicated that gut microbiota markers such as diversity, f/b ratios, and gut richness/abundance differ between patients with disease and those without disease. Other included studies also demonstrated differential taxonomic profiles between disease and non-disease states. Moreover, some of the clinical trials showed that there was improvement in the gut, which was demonstrated by restored/increased diversity and the presence of certain beneficial/protective gut microbes after the administration of certain interventions. However, as expected with gut microbiota studies, some authors reported contrasting results (Table 2).

## 4. Discussion

The analysis of gut microbiota has gained significant attention in recent years, as mounting evidence suggests its crucial role in human health and disease. This scoping review plays a vital role in summarizing the breadth and depth of existing gut microbiota research in Africa, therefore providing a comprehensive overview of the available literature. Overall, our results indicate a comprehensive coverage of the literature published between December 2010 and June 2023, including a mix of scholarly articles and posters/abstracts. The varying sample sizes (3–1900) across the studies reflect the diversity in research approaches and highlight the need for careful consideration of the context and goals of each study when interpreting the findings.

When looking at the study groups, it is evident that children were the most studied group, indicating a significant focus on research involving children. The reasons for this could be attributed to the importance of understanding child development, health, and wellbeing. The second most studied population group was adults. This is not surprising, as adults make up a sizable portion of the general African population [180], and are often the target of various health and social interventions. This demonstrates the need to shift the focus more to adults, as this population is more prone to acquiring many of the diseases of public health significance, such as diabetes mellitus [181], stroke, and ischaemic heart disease [182], as well as other lifestyle diseases.

A significant majority of the studies (90.3%) did not mention the specific ethnic group or tribe that the research was conducted within. This indicates a potential gap in reporting this important demographic information, which can be crucial for understanding health disparities and the impact of interventions on specific populations. Furthermore, notable gut microbiota discrepancies have been observed among individuals belonging to the same community, and these variations were linked to variances in their ethnic backgrounds [183]. Gender is another important variable to include in gut microbiota studies [184]. Interestingly, more than a quarter (*n* = 48) of the included studies did not mention the number of males and females included in the research. This lack of reporting makes it challenging to draw concrete conclusions about the gender distribution in these studies. The age of the populations studied in the research studies included varied widely, ranging from 0 months (newborns) to 84 years. This broad age range suggests a diverse focus, spanning from early developmental stages to older adulthood.

The scoping review revealed that a considerable proportion of the studies included (50%) did not specify the type of study setting in which their research was conducted. The lack of information regarding the study settings indicates a potential gap in reporting, which can limit the understanding of the context in which the research took place. The absence of this crucial detail makes it challenging to assess the generalizability and applicability of the study findings to specific populations or settings. Moreover, the omission of this information restricts our ability to fully comprehend the contextual factors that may influence gut microbiota research outcomes. However, among the studies that did indicate the study setting, most of them (27.9%) were conducted in rural areas. This finding suggests a focus on examining diseases or interventions (in relation to gut microbiota) in rural communities. The emphasis on rural areas could be driven by a range of factors, such as the unique challenges faced by these communities, differences in health outcomes, or the need to address health disparities between rural and urban populations [185]. 

In addition, the current review revealed a limited number of studies investigating diseases of public health significance in relation to the gut microbiota. Among these studies, HIV, colorectal cancer, and diabetes mellitus received the most attention. The distribution of disease focus varied across countries, with some countries placing a higher emphasis on specific diseases. For instance, South Africa predominantly studied HIV (40.9%), while colorectal cancer was primarily studied in Egypt (25%), and diabetes mellitus was studied mostly in Egypt (50%) as well. Surprisingly, a considerable number of studies (42.9%) in this review focused more broadly on gut microbiota without specifically examining their links to diseases. 

Investigating the gut microbiota in relation to disease is essential for gaining insights into disease mechanisms [186], identifying biomarkers [187], developing targeted therapies [188], and advancing precision medicine [189]. However, on the other hand, the considerable proportion of studies focusing on the broader aspects of gut microbiota composition, functions, or associations rather than specific diseases reflects the growing interest in understanding the complexities of this microbial community. These studies contribute to foundational knowledge and pave the way for future disease-specific investigations. By unravelling the complexities of the gut microbiota, researchers can gain a deeper understanding of its potential impacts on health and explore novel therapeutic avenues [190].

Looking at the distribution of countries in which gut microbiota research was conducted, the results indicate that several countries, including South Africa, Malawi, Egypt, Kenya, and Nigeria, have made noteworthy contributions to gut microbiota research. These countries have demonstrated research expertise, active research communities, and ongoing efforts to understand the role of the gut microbiota in health and disease. The collective efforts of these countries contribute to the global knowledge base and help advance our understanding of the gut microbiota’s impact on human health. However, the high number (*n* = 66) of studies that did not study any specific disease in relation to the gut microbiota remains a gap that needs to be filled in African countries.

Moreover, it is of concern to see the relatively small number of gut microbiota research publications from the African continent as compared to Western countries. This demonstrates the level of under-representation of African populations [191]. The under-representation of gut microbiota research in Africa can be attributed to several factors. One primary factor is the limited research infrastructure and resources in many African countries [192]. The lack of funding, laboratory facilities, and trained personnel can hinder the conduct of extensive research in the gut microbiota field [192]. Additionally, there may be a lack of awareness or recognition of the importance of gut microbiota research among policymakers, healthcare providers, and researchers in some African countries. Limited attention and investment in this area of research can result in a lack of research initiatives and a lower overall output.

When evaluating the methodologies used, our results demonstrate an increased preference for 16S rRNA/meta-genomic sequencing, followed by quantitative PCR/real-time PCR as the methods of choice for gut microbiota profiling. These methods offer insights into the composition and abundance of microbial taxa, contributing to our understanding of the gut microbiota’s role in health and disease. However, there are other emerging methods that require attention, such as metatranscriptomics and metabolomics. These methods are important because, as much as metagenomics provides information about the taxonomic composition of the sample, metatranscriptomics allows us to understand the functional profile and metabolomics reveals the byproducts released into the environment, thereby completing the comprehensive picture [193]. 

Regarding the sample types, the selection of the type of sample used for gut microbiota profiling is crucial [194] because this can determine the quality and accuracy of the sequencing results. Stool samples were used for gut microbiota profiling in most of the included studies. They are commonly employed in gut microbiota research [195] due to their non-invasive collection method [196]. Stool samples provide insights into the overall composition and diversity of the gut microbiota and are relatively easy to collect from study participants. Interestingly, some studies used rectal swabs, although the collection procedure comes with more discomfort than stool sampling and potential contamination with skin bacteria can occur [196], while some opted for colonic mucosal tissues, which is an even more invasive process that is relatively costly. 

The current scoping review documented generalized key findings of all the included studies, as shown in Table 2. The collective results of the review suggest that gut microbiota markers and taxonomic profiles differ between disease and non-disease states [190], indicating a potential association between the gut microbiota and various health conditions. However, it is of note to be aware that the methodologies used to identify components of the gut microbiota vary across studies, leading to inconsistencies and challenges when comparing and establishing reference standards [197] (specific gut microbiota traits). Additionally, the results highlight the potential for interventions to positively impact the composition and diversity of the gut microbiota. However, it is crucial to acknowledge the existence of contrasting results in gut microbiota research [198], emphasizing the need for further research to elucidate the mechanisms underlying these associations and to address the sources of variability across studies. 

One crucial aspect of this scoping review is examining the diverse types of studies employed in the included studies. This is important because the validity and generalizability of research findings greatly depend on the suitability of the selected study design. The results of the current scoping review indicate that observational/cross-sectional studies are the most prevalent study types, followed by experimental studies/clinical trials, case–control studies, sub-studies, and comparative studies. Observational studies such as cross-sectional studies can help establish correlations between microbial composition and various factors such as demographics, lifestyle, and disease status. Nonetheless, cross-sectional studies encounter two primary drawbacks: they often struggle to establish a definitive cause-and-effect relationship and are susceptible to survival bias [199]. 

Other observational studies that are predominant in the included studies are case–control studies. The advantages of such study types lie in their feasibility in resource-limited settings and their capacity to examine numerous exposures related to a specific outcome [199]. Nevertheless, case–control studies come with certain drawbacks. These include the potential for cases to be influenced by recall bias [199]. Additionally, as case–control studies do not start with a defined population at risk, they cannot establish the actual risk of the outcome [199]. Instead, researchers can only calculate the odds of association between exposure and the outcome [199]. 

Comparative studies were also conducted in several of the included studies. These studies encompass a wide range of design options, which include experimental versus observational as well as prospective versus retrospective approaches [200]. As a result, the quality of comparative studies is dependent on several methodological factors, such as the selection of variables, sample size, consideration of bias sources, confounders, and adherence to quality and reporting standards [200]. The use of the previously mentioned observational study types in most of the included studies is accepted; however, more experimental studies such as clinical trials are required [188] to provide valuable insights into the cause-and-effect relationships between interventions, the gut microbiota, and disease-related parameters. In addition to these trials being expensive to conduct and resource intensive [199], they are considered to be the gold-standard design [199,201].

## 5. Strengths and Limitations

This scoping review allowed for a broad and inclusive assessment of existing research on the gut microbiota in the African context. This approach ensured that a wide range of sources and perspectives were considered, therefore providing a comprehensive overview of the subject. Furthermore, the review effectively identified gaps in the current state of gut microbiota research in Africa. This insight can guide future research endeavours and help prioritize areas that require further investigation. Lastly, the review findings could have practical implications for public health policies and clinical practices in the region. Understanding the existing knowledge base regarding gut microbiota in Africa may inform strategies to improve health outcomes and address specific health challenges. 

However, the current scoping review might be limited by a language bias, as it only included research published in English. This could potentially omit valuable contributions published in other languages, leading to an incomplete representation of the subject matter. The scoping review additionally excluded research that examined both animals and humans. Despite the limited number of such studies identified, this exclusion might lead to a potential oversight of useful human data concerning the gut microbiota. Finally, the review did not thoroughly assess the methodological rigor of individual studies, which might potentially affect the reliability of the synthesized information.

## 6. Conclusions

Overall, this scoping review provides valuable insights into gut microbiota research in Africa, highlighting the associations between gut microbiota markers, taxonomic profiles, and health conditions. It also underscores the potential for interventions to positively impact the gut microbiota composition and diversity. However, the presence of contrasting results emphasizes the need for further research to elucidate underlying mechanisms and address sources of variability. The review also reveals a limited number of studies investigating diseases of public health significance in relation to the gut microbiota, with variations in disease focus across countries. This highlights the importance of further research to explore the associations between gut microbiota and specific diseases (especially those of public health significance), as it can provide insights into disease mechanisms, biomarkers, and therapeutic strategies to improve human health. Additionally, the scoping review emphasizes the significance of incorporating additional experimental studies to establish definitive cause-and-effect relationships between the gut microbiota and disease-related parameters.

## Figures and Tables

**Figure 1 microorganisms-11-02118-f001:**
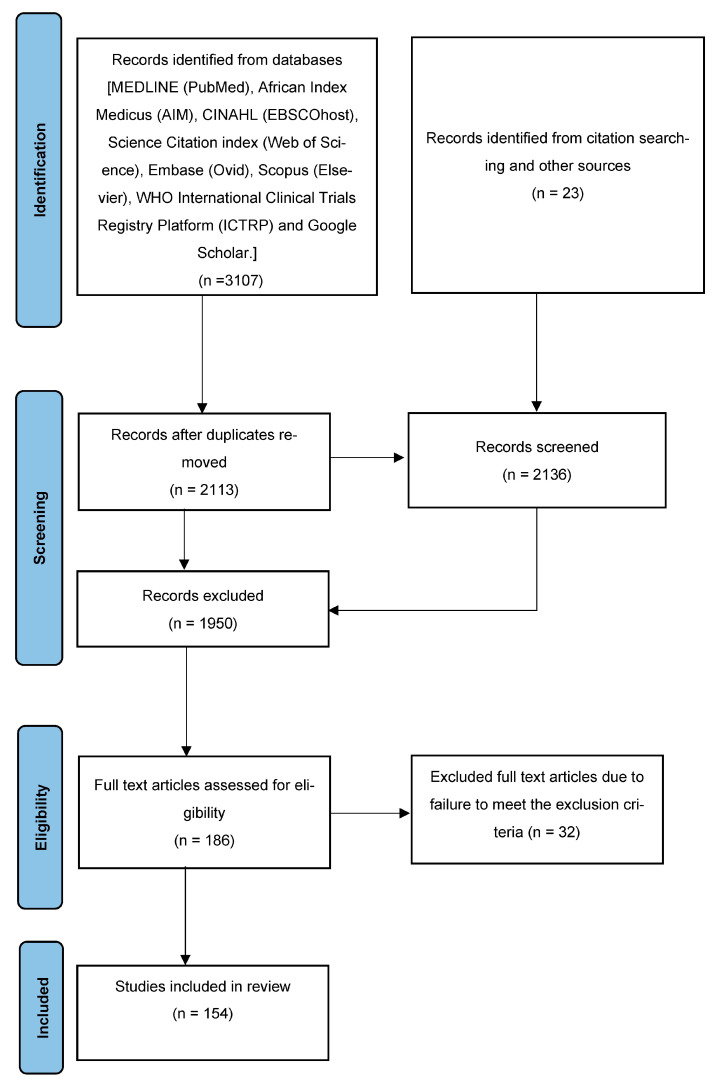
Flow diagram [25] outlining the selection process.

**Figure 2 microorganisms-11-02118-f002:**
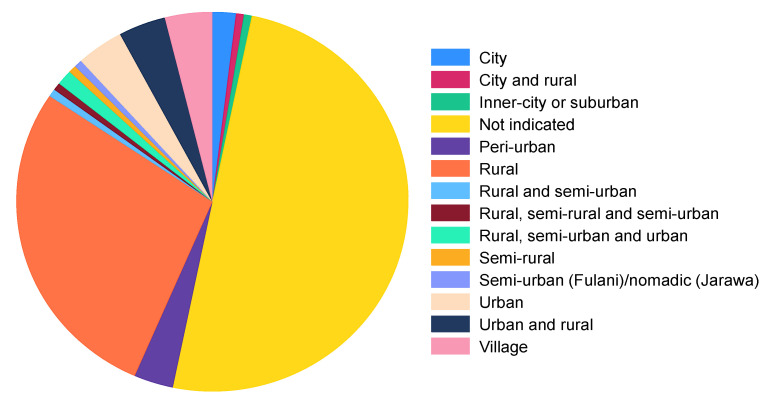
Distribution of study settings.

**Figure 3 microorganisms-11-02118-f003:**
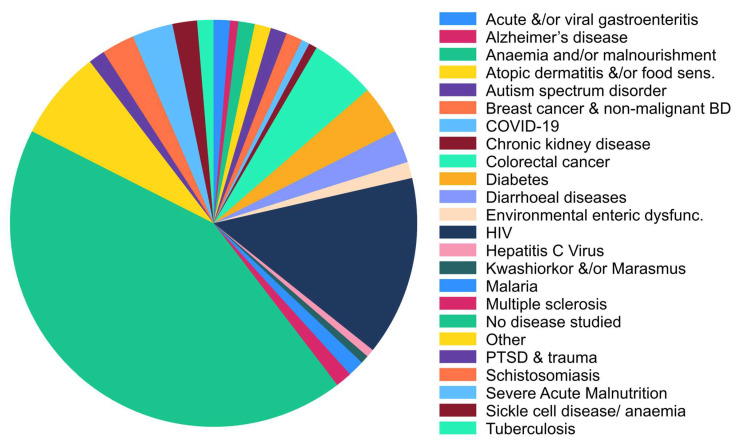
Distribution of different diseases studied in relation to the gut microbiota. BD = breast disease. Other: a combination of distinct types of diseases, and those that appeared once and are also not of public health significance.

**Figure 4 microorganisms-11-02118-f004:**
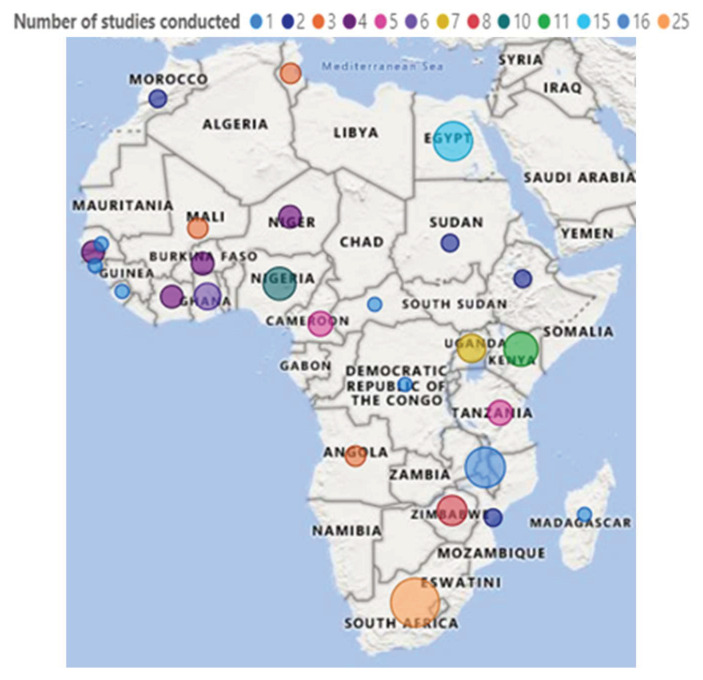
Distribution of countries. Other, *n* = 6 (3.9%): a combination of different countries.

**Figure 5 microorganisms-11-02118-f005:**
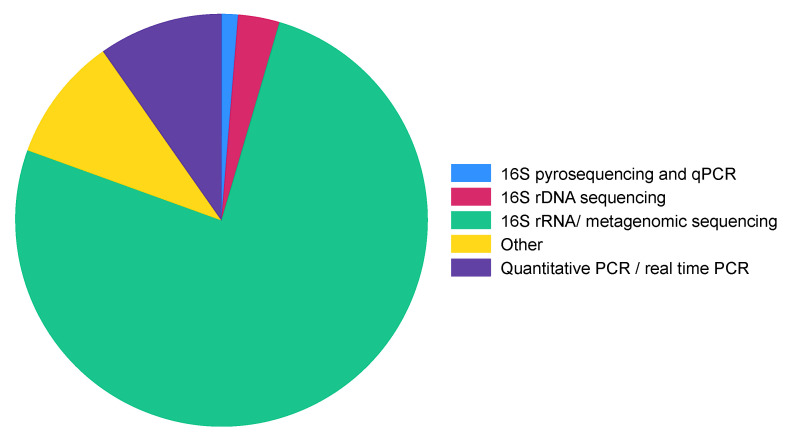
Distribution of method types that were used for gut microbiota profiling. Other: a combination of more than one type of method used, and method types that appeared once.

**Figure 6 microorganisms-11-02118-f006:**
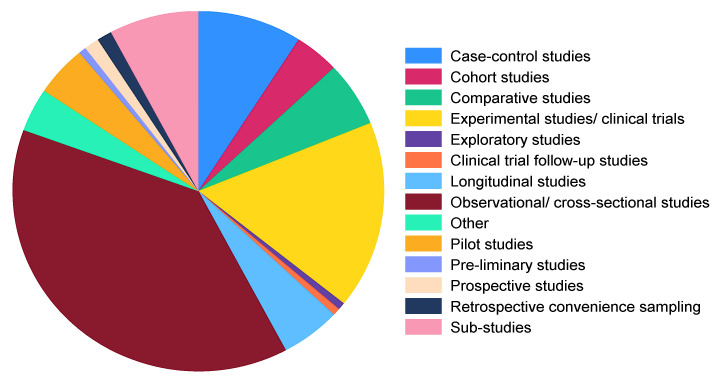
Distribution of study types/designs employed in the included studies. Other: a combination of more than one study type/design.

**Figure 7 microorganisms-11-02118-f007:**
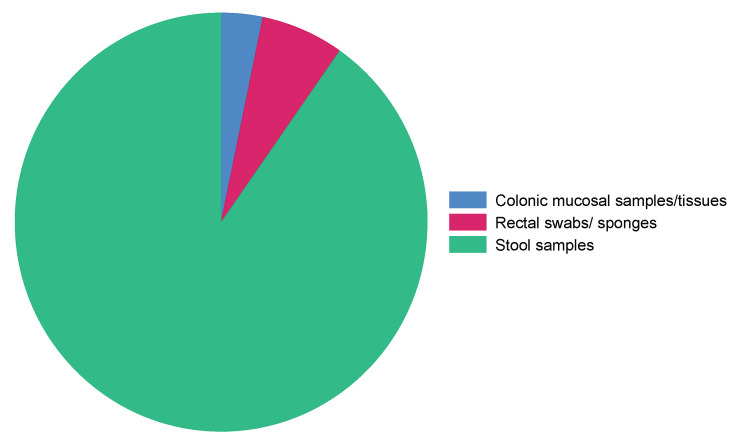
Distribution of sample types utilized for gut microbiota profiling.

**Table 1 microorganisms-11-02118-t001:** Population characteristics.

Population Group	Percentage (%)
Adults	27.9
Children	37.1
Infants	10.4
Not indicated	2.0
Other	22.7
**Ethnicity/Tribe**	**Percentage (%)**
Bambara	0.7
Black	2.6
Hadza	0.7
Mixed ancestry	1.3
Not indicated	90.3
Other	4.6
**Gender Ratio (Female/Male) Groups**	** *n* **
Ratios between 0.31–0.99	35
Ratios equal to 1	9
Ratios between 1.05–5.83	52
Females only	7
Males only	3
Not indicated	48
**Age range:**	0 months–84 years

Other in population group: a combination of different population groups. Other in ethnic groups: a combination of different ethnic groups/tribes.

**Table 2 microorganisms-11-02118-t002:** General/key findings of the included studies.

Study Title	Author’s Name and Year of Publication	General/Key Findings of the Studies	Type of Study
Provision of lipid-based nutrient supplements to mothers during pregnancy and 6 months postpartum and to their infants from 6 to 18 months promotes infant gut microbiota diversity at 18 months of age but not microbiota maturation in a rural Malawian setting: secondary outcomes of a randomised trial	Kamng’Ona et al., 2020 [27]	The findings “did not support the hypothesis that LNS supplementation will promote gut microbiota maturity in Malawian infants”.	Clinical trial
Infant gut microbiota characteristics generally do not modify effects of lipid-based nutrient supplementation on growth or inflammation: Secondary analysis of a randomised controlled trial in Malawi	Hughes et al., 2020 [28]	“No conclusive evidence of effect modification was observed in this analysis, the relationships observed before correction for multiple hypothesis testing may be worth additional investigation”.	Clinical trial
The gut microbiome and early-life growth in a population with high prevalence of stunting	Robertson et al., 2023 [29]	“HIV exposure shapes maturation of the infant gut microbiota, and the functional composition of the infant gut microbiome is moderately predictive of infant growth in a population at high risk of stunting”.	Sub-study of a clinical trial
The impact of anthelmintic treatment on human gut microbiota based on cross-sectional and pre- and post-deworming comparisons in western Kenya	Easton et al., 2019 [30]	The authors were able to “identify changes in the microbiota associated with clearance of *N. americanus* infection, which were not seen posttreatment in individuals who were uninfected pretreatment”.	Longitudinal and cross-sectional study
Alteration of the gut fecal microbiome in children living with HIV on antiretroviral therapy in Yaounde, Cameroon	Abange et al., 2021 [31]	“HIV-infected, antiretroviral therapy (ART) -treated children were characterized by decreased alpha diversity and shifts in community structure. ART regimen was associated with varying degrees of dysbiosis with ritonavir-boosted protease inhibitor (PI/r) based regimens”.	Case-control
Evolution in fecal bacterial/viral composition in infants of two central African countries (Gabon and Republic of the Congo) during their first month of life	Brazier et al., 2017 [32]	“The bacterial microbiota communities displayed a similar diversification and expansion in newborns within and between countries during the first four weeks of life”.	Longitudinal study
Anaerobe-enriched gut microbiota predicts pro-inflammatory responses in pulmonary tuberculosis	Naidoo et al., 2021 [33]	“Specific anaerobes in cases’ stool predict upregulation of pro-inflammatory immunological pathways, supporting the gut microbiota’s role in TB”.	Cross-sectional study
A comparative study of the gut microbiome in Egyptian patients with type I and type II diabetes	Radwan et al., 2020 [34]	The results highlighted “a significant increase in abundance of Gram negative, potentially opportunistic pathogenic taxa (*Pseudomonas*, *Prevotella*) in all diabetic groups. The gram-positive *Gemella*, also had a significant increase in abundance in all diabetic groups. *Turicibacter*, *Terrisporobacter* and *Clostridium* were found to be more abundant in the control group than in type I diabetes (TID)”.	Comparative study
Investigations on the interplays between *Schistosoma mansoni*, praziquantel and the gut microbiome	Schneeberger et al., 2018 [35]	“Overall taxonomic profiling and diversity indicators were found to be close to a “healthy” gut structure in all children. Slight overall compositional changes were observed between *S. mansoni*-infected and non-infected children. Praziquantel treatment was not linked to a major shift in the gut taxonomic profiles”.	Observational study
Maternal human milk oligosaccharide profile modulates the impact of an intervention with iron and Galacto-oligosaccharides in Kenyan infants	Paganini et al., 2019 [36]	“Infants of non-secretor mothers may be more vulnerable to the adverse effect of fortificant iron on the gut microbiota, resulting in decreased abundances of *Bifidobacterium* and increased abundances of enteropathogens, but also benefit more from the co-provision of Galacto-oligosaccharides in terms of beneficial effects on the gut microbiota and improving iron status”.	Sub-study of a clinical trial
Iron-containing micronutrient powders modify the effect of oral antibiotics on the infant gut microbiome and increase post-antibiotic diarrhoea risk: a controlled study in Kenya	Paganini et al., 2019 [37]	“In African infants, iron fortification modifies the response to broad-spectrum antibiotics: iron may reduce their efficacy against potential enteropathogens, particularly pathogenic *E. coli*, and may increase risk for diarrhoea”.	Clinical trial
Design and application of a novel two-amplicon approach for defining eukaryotic microbiota	Popovic et al., 2018 [38]	“The combined sequence information allowed the authors to uncover protozoa, microsporidia, helminths, and fungi, and putative relationships between the eukaryote *Blastocystis* and bacteria”.	Observational study
Brief overview of dietary intake, some types of gut microbiota, metabolic markers and research opportunities in sample of Egyptian women	Hassan et al., 2022 [39]	“Dietary factors, dysbiosis, and the metabolic product short chain fatty acids have been implicated in causing metabolic defects”.	Cross-sectional study
Microbiota at multiple body sites during pregnancy in a rural Tanzanian population and effects of moringa-supplemented probiotic yogurt	Bisanz et al., 2015 [40]	“Microbiota analysis by weighted UniFrac distances comparing samples to enrolment showed that moringa-supplemented probiotic yogurt does not affect the microbiota structure and that the faecal microbiotas remained stable over pregnancy”.	Clinical trial
Randomised open-label pilot study of the influence of probiotics and the gut microbiome on toxic metal levels in Tanzanian pregnant women and school children	Bisanz et al., 2014 [41]	“Administration of the probiotic was not observed to have an effect on the gut bacterial community composition. Elevated blood lead was associated with increases in *Succinivibrionaceae* and *Gammaproteobacteria* relative abundance levels in stool”.	Pilot clinical trial
The microbiome in posttraumatic stress disorder and trauma-exposed controls: An exploratory study	Hemmings et al., 2017 [42]	“Measures of overall microbial diversity were similar among individuals with post-traumatic stress disorder (PTSD) and Trauma-exposed (TE) controls; however, decreased total abundance of *Actinobacteria*, *Lentisphaerae*, and *Verrucomicrobia* was associated with PTSD status”.	Cross-sectional
Early benefits of a starter formula enriched in prebiotics and probiotics on the gut microbiota of healthy infants born to HIV+ mothers: A randomised double-blind controlled trial	Cooper et al., 2016 [43]	“The bovine milk-derived oligosaccharides (BMOS) prebiotic in combination with *B. lactis* probiotic stimulated the growth of bifidobacteria in infants born by cesarean delivery at early life (within the first 10 days) when the gut colonization with bifidobacteria is delayed compared to vaginally born infants”.	Clinical trial
The gut microbiome in konzo	Bramble et al., 2021 [44]	“Gut microbiome structure is highly variable depending on region of sampling, but most interestingly, the authors identify unique enrichments of bacterial species and functional pathways that potentially modulate the susceptibility of konzo in prone regions of the Congo”.	Observational study
Study of gut microbiome in Egyptian patients with autoimmune thyroid diseases	El-Zawawy et al., 2021 [45]	“Egyptian patients with autoimmune thyroid disorders (ATD) (Graves’ disease (GD) and Hashimoto’s thyroiditis (HT)) show dysbiosis of the gut microbiome”.	Observational study
Gut microbiome alterations in patients with stage 4 hepatitis C	Aly et al., 2016 [46]	“The alpha diversity of the healthy persons’ gut microbiomes was higher than those of the hepatitis C virus (HCV) patients. Patients with HCV had a few significant fecal microbiome changes”.	Case-control study
Restitution of gut microbiota in Ugandan children administered with probiotics (*Lactobacillusrhamnosus GG* and *Bifidobacterium animalis* subsp. *lactis BB-12*) during treatment for severe acute malnutrition	Castro-Mejía et al., 2020 [47]	“Children with severe acute malnutrition (SAM) have significantly reduced number of observed species and major compositional differences (β-diversity) compared to healthy subjects. Moreover, gut microbiota (GM) diversity and composition change over the course of rehabilitation from SAM and approach the GM of apparently healthy subjects as treatment progresses”.	Sub-study of a clinical trial
Characterization and profiling of gut bacterial microbiome and pathobionts among HIV-negative and HIV-infected individuals in Cameroon	Eyongabane Ako et al., 2022 [48]	“Gut pathobionts are circulating among HIV-infected and HIV-negative individuals in Cameroon. Unique gut microbiome OTU (operational taxonomic unit) sequences are significantly high among HIV-infected. Emerging strains of new microorganisms are on the rise”.	Case-control and comparative study
The role of probiotics in children with autism spectrum disorder: A prospective, open-label study	Shaaban et al., 2018 [49]	“Probiotics have beneficial effects on both behavioral and gastrointestinal (GI) manifestations of autism spectrum disorder (ASD)”.	Clinical trial
Metagenomic analysis of gut microbiota of patients with colorectal cancer at the Federal Medical Centre (FMC), Abeokuta, Ogun State, Nigeria	Sulaimon et al., 2023 [50]	“The taxonomic composition and functional genes of intestinal bacteria were significantly altered in colorectal cancer (CRC). Also, *E. coli* and *P. aeruginosa* are at least partially involved in the pathogenesis of CRC”.	Observational study
Alteration of gut microbiota in Alzheimer’s disease and their relation to the cognitive impairment	Khedr et al., 2022 [51]	“The current work highlighted a significant relationship between Alzheimer’s disease (AD) and gut microbiota dysbiosis”.	Case-control study
Study of the gut microbiome profile in children with autism spectrum disorder: A single tertiary hospital experience	Ahmed et al., 2020 [52]	“The current study showed evidence of changes in the gut microbiome of autism spectrum disorder (ASD) children compared to the unrelated controls. However, the microbiome profile of siblings was more like that of autistic children than that of unrelated controls”.	Observational study
Molecular characterization of the gut microbiome in Egyptian patients with remitting relapsing multiple sclerosis	Mekky et al., 2022 [53]	“Egyptian patients with multiple sclerosis exhibit microbial dysbiosis. Multiple sclerosis patients have significantly higher *B. fragilis.* The level of *Prevotella*, *Lactobacilli* and *C. perfringes* appear much less in MS patients than the control”.	Observational study
Dysbiosis signatures of fecal microbiota in South African infants with respiratory, gastrointestinal, and other diseases	Krishnamoorthy et al., 2020 [54]	“The authors showed potential links between the fecal microbiota and clinical parameters, disease-based signature microbiota, and the marker pathogens”.	Case-control study
The effects of iron fortification on the gut microbiota in African children: a randomised controlled trial in Cote d’Ivoire	Zimmermann et al., 2010 [55]	“Anaemic African children carry an unfavourable ratio of fecal enterobacteria to bifidobacteria and *lactobacilli*, which is increased by iron fortification. Thus, iron fortification in this population produces a potentially more pathogenic gut microbiota profile, and this profile is associated with increased gut inflammation”.	Clinical trial
Lipid-based nutrient supplements do not affect gut *Bifidobacterium* microbiota in Malawian infants: A randomised trial	Aakko et al., 2017 [56]	“The dietary supplementation did not have an effect on the *Bifidobacterium* and *S. aureus* microbiota composition of the study infants. The fecal bifidobacterial diversity of the infants, however, changed toward a more adult-like microbiota profile within the observed time”.	Clinical trial
Associations between HIV status and the gut microbiota in South African children with low iron stores	Goosen et al., 2021 [57]	“*Prevotella*-enrichment, evident in both groups, was likely influenced by the plant-based diet. The significantly lower relative abundance of beneficial *Bifidobacterium* among the HIV+ children may be cause for concern as reductions in *Bifidobacterium* following oral iron supplementation have been reported”.	Comparative cross-sectional study
Gut microbiome 16S rRNA gene amplicon taxonomic profiling of hospitalized Moroccan COVID-19 patients	Sehli et al., 2022 [58]	“The 16S rRNA gene meta-taxonomic profiling method revealed differences in microbiome composition and richness changes between hospitalized/treated COVID-19 patients and healthy controls”.	Comparative study
Metabolic derangements identified through untargeted metabolomics in a cross-sectional study of Nigerian children with severe acute malnutrition	McMillan et al., 2017 [59]	“The plasma metabolome discriminated children with SAM from controls, while no significant differences were observed in the microbial or small molecule composition of stool”.	Cross-sectional study
Parasites and diet as main drivers of the Malagasy gut microbiome richness and function	Mondot et al., 2021 [60]	“High protozoan carriage was associated with higher diversity, richness, and microbial functionalities. Asymptomatic protozoan carriage and dietary habits are the external factors with the deepest impact on gut microbiome”.	Observational study
Metagenomic profiling of gut microbiota of diarrhoeic children in Southwest Nigeria	Ugboko et al., 2020 [61]	In diarrheal samples, *Firmicutes*, *Proteobacteria*, *Actinobacteria*, *Bacteroidetes*, and *Fusobacteria* were prominent, except *Verrucomicrobia*. *Proteobacteria* were notably reduced in controls, with heightened species richness (*Escherichia coli, Shigella*, etc.) in the diarrhoeic samples, and increased *Bifidobacterium, Faeacalibacterium*, etc., in controls.	Observational study
Evaluating the gut microbiome in children with stunting: Findings from a South African birth cohort	Budree et al., 2019 [62]	“The findings demonstrate a microbial signature associated with stunting in African children”.	Observational study
Microbiota richness and diversity in a cohort of underweight HIV positive children aged 24–72 months in Cape Town, South Africa	Van Niekerk et al., 2019 [63]	“Diminished growth of *Clostridium Perfringes* and increased growth of Enterobacteria was shown, the cohort had low diversity of microbiota. *Firmicutes* phyla was reasonably well represented in the cohort”.	Observational study
The gut microbiota’s influence in the development of foetal alcohol spectrum disorders	Kitchin et al., 2021 [64]	“There were no significant differences in alpha- or beta-diversity, however Bristol Stool Scale and delivery mode was shown to influence beta diversity. Bifidobacteria and *Prevotella* were found to be higher in infants diagnosed with foetal alcohol spectrum disorder (FASD)”.	Observational study
Gut microbiota related to *Giardia duodenalis, Entamoeba* spp. and *Blastocystis hominis* infections in humans from Côte d’Ivoire	Iebba et al., 2016 [65]	“This preliminary investigation demonstrates a differential fecal microbiota structure in subjects infected with *G. duodenalis* or *Entamoeba* spp./*B. hominis*”.	Pre-liminary study
Frequency of *Firmicutes* and *Bacteroidetes* in gut microbiota in obese and normal weight Egyptian children and adults	Ismail et al., 2011 [66]	“Obesity in Egyptian children and adults is associated with compositional changes in faecal microbiota with increase in the phyla *Firmicutes* and *Bacteroidetes*”.	Observational study
Gut microbiome community composition is significantly influenced by shared living- space in rural agriculturalists from Burkina Faso	Jacobson et al., 2019 [67]	“Intra-village gut microbiome variation is driven primarily by sharing or lack thereof of quartiers, providing a link between microbial ecological dynamics and living space”.	Observational study
Molecular characterization of gut microbial structure and diversity associated with colorectal cancer patients in Egypt	El-Sokkary, 2022 [68]	“The results demonstrated increased abundance of *Fusobacterium* or *Bifidobacterium*, and that they can be considered as a sign for impairment or a diseased condition”.	Observational study
A prospective study on child morbidity and gut microbiota in rural Malawi	Kortekangas et al., 2019 [69]	“Specific morbidity symptoms might be associated with changes in the relative abundances of several bacterial taxa and overall microbial community composition. There was no clear consistent pattern in the associations between microbiota and morbidity”.	Prospective study
Gut bacterial diversity and growth among preschool children in Burkina Faso	Digitale et al., 2020 [70]	The authors “did not find evidence that gut microbial diversity was associated with growth”.	Sub-study of a clinical trial
Longitudinal analysis of infant stool bacteria communities before and after acute febrile malaria and artemether-lumefantrine treatment	Mandal et al., 2019 [71]	“In-depth bioinformatics analysis of stool bacteria has revealed for the first time that human malaria episode/artemether-lumefantrine (AL) treatment have minimal effects on gut microbiota in Kenyan infants”.	Longitudinal study
Infant and adult gut microbiome and metabolome in rural Bassa and urban settlers from Nigeria	Ayeni et al., 2018 [72]	“The data highlight specific microbiome traits that are progressively lost with urbanization, such as the dominance of pristine fibre degraders and the low inter-individual variation”.	Cross-sectional study
New insights on obesity and diabetes from gut microbiome alterations in Egyptian adults	Salah et al., 2019 [73]	“Both obesity and diabetes greatly affect and are affected by gut microbiota. The relative abundance of *Firmicutes* and *Bacteroidetes* phyla in obese and diabetic individuals and the F/B ratio are still the most discriminative factors between the patient cases and healthy individuals”.	Comparative study
Composition of gut microbiota of children and adolescents with perinatal Human immunodeficiency virus infection taking antiretroviral therapy in Zimbabwe	Flygel et al., 2020 [74]	“Gut microbiota is altered in HIV-infected children, although diversity improves with increasing duration of ART”.	Cross-sectional study
Association of maternal prenatal psychological stressors and distress with maternal and early infant faecal bacterial profile	Naudé et al., 2020 [75]	“Maternal prenatal exposure to intimate partner violence is associated with differences in faecal bacterial profiles of mothers at delivery (*Lactobacillaceae* and *Peptostreptococcaceae*) and of their infants at birth (family *Enterobacteriaceae*) and 4–12 weeks of age (genus *Weissella*). Higher psychological distress during pregnancy is associated with lower infant faecal bacterial profiles of the family *Veillonellaceae* at 20–28 weeks”.	Longitudinal study
Differences in the faecal microbiome in *Schistosoma haematobium* Infected children vs. uninfected children	Kay et al., 2015 [76]	“There are significant differences in the gut microbiome structure of infected vs. uninfected children and the differences were refractory to antihelminthic praziquantel (PZQ) treatment”.	Cross-sectional study
Urogenital schistosomiasis is associated with signatures of microbiome dysbiosis in Nigerian adolescents	Ajibola et al., 2019 [77]	“The adolescent gut microbiome may be shifted towards a dysbiotic state by infection with *S. haematobium*”.	Observational study
Gut microbiome profiles are associated with type 2 diabetes in urban Africans	Doumatey et al., 2020 [78]	“Non-type 2 diabetes (T2D) adults living in a Nigerian city have a characteristic microbial composition that is mainly composed of *Firmicutes (Clostridiales)* and *Actinobacteria* (90%). The GM of cases have a bacterial signature consisting of increased sulfate-reducing spp. *Desulfovibrio piger, Prevotella*, *Peptostreptococcus*, and *Eubacterium* and is characterized by a moderate dysbiosis which features a decrease in *Firmicutes*”.	Case-control study
Analysis of human gut microbiota composition associated to the presence of commensal and pathogen microorganisms in Côte d’Ivoire	Di Cristanziano et al., 2021 [79]	“The co-occurrence of *Blastocystis* and commensal *Entamoeba* spp., despite the presence of other enteric pathogens including *G. duodenalis*, seems to preserve a high diversity, favour different bacterial consortia, and does not compromise the intrinsic ability of intestinal microbiota to restore and/or maintain homeostasis”.	Observational study
Effects of iron supplementation on dominant bacterial groups in the gut, faecal SCFA and gut inflammation: A randomised, placebo-controlled intervention trial in South African children	Dostal et al., 2014 [80]	“The present study suggests that in African children with a low enteropathogen burden, iron (Fe) status and dietary Fe supplementation did not significantly affect the dominant bacterial groups in the gut, faecal short chain fatty acid (SCFA) concentration or gut inflammation”.	Clinical trial
Rectal microbiota among HIV-uninfected, untreated HIV, and treated HIV-infected in Nigeria	Nowak et al., 2017 [81]	“There were subtle shifts in the rectal microbiota among HIV-positive individuals receiving treatment, including a decrease in diversity of the *Bacteroidetes* phylum driven primarily by a loss in *Prevotella*. There was a shift towards more pathogenic bacteria. Untreated HIV infection does not significantly alter the rectal microbiota, whereas prior treatment is associated with a shift toward a more pathogenic pattern of microbiota”.	Cross-sectional study
*Mycobacterium avium* subsp. *paratuberculosis* and microbiome profile of patients in a referral gastrointestinal diseases centre in the Sudan	Elmagzoub et al., 2022 [82]	“A unique microbiome profile of *Mycobacterium avium* subsp. *paratuberculosis* (MAP) -positive patients in comparison to MAP-negative was found”.	Cross-sectional study
Absence of *Mitsuokella multacida* is associated with early onset of colorectal cancer	Elkholy et al., 2020 [83]	“These findings suggest that the oncoprotective effect of *Mitsuokella multacida* should be further investigated”.	Retrospective convenience sample design
Profile of microbiota is associated with early onset of colorectal cancer in Egyptian and Kenyan patients	Arafat, 2020 [84]	“The findings suggest that the oncoprotective effect of *Mitsuokella multacida* should be further investigated”.	Retrospective convenience sample de-sign
Gut microbiota in Malawian infants in a nutritional supplementation trial	Cheung et al., 2016 [85]	“Infants who received iron-containing LNS or corn–soya blend (CSB) for 12 months did not differ from non-recipients in the gut microbiota profiles”.	Sub-study of a clinical trial
Altered faecal microbiota composition and structure of Ghanaian children with acute gastroenteritis	Quaye et al., 2023 [86]	“The faecal microbiota of acute gastroenteritis (AGE) cases was dominated by disease-associated bacterial genera. Finally, whole microbial community network characteristics differed between AGE cases and controls”.	Cross-sectional case control study
The association of gut microbiota characteristics in Malawian infants with growth and inflammation	Kamng’ona et al., 2019 [87]	“Microbiota diversity and maturity were related to growth in weight from 6 to 12 months, but not to growth in length or head circumference or to growth from 12 to 18 months. Microbiota diversity and maturity may also be linked to inflammation, but findings were inconsistent”.	Longitudinal study
Maternal fecal microbiome predicts gestational age, birth weight and neonatal growth in rural Zimbabwe	Gough et al., 2021 [88]	“The pregnancy faecal microbiome, primarily the abundance of resistant-starch degraders, is an important contributor to birth weight and neonatal growth, and to a lesser extent gestational age, in infants of rural Zimbabwean mothers who consume a diet high in maize”.	Sub-study of a clinical trial
Altered virome and bacterial microbiome in human immunodeficiency virus-associated acquired immunodeficiency syndrome	Monaco et al., 2016 [89]	“Severe immunodeficiency is likely the mechanism leading to changes in the fecal microbiome, including bacteria and viruses. Immune reconstitution, such as through early ART, may restore the healthy enteric microbiome”.	Observational study
Changes in the gut microbiota of Nigerian infants within the first year of life	Oyedemi et al., 2022 [90]	“The observed taxonomic differences in the gut microbiota between pre-weaning and weaning samples in Nigerian infants, as well as butyrate production, were influenced by diet. Introduction of solid foods encouraged an increase in microbial diversity, helpful for a healthy life”.	Longitudinal cohort study
Child development, growth and microbiota: follow-up of a randomised education trial in Uganda	Atukunda et al., 2019 [91]	“The maternal education intervention had positive effects on child development and growth at three years but did not alter gut microbiota composition”.	Follow-up study of a clinical trial
Early antiretroviral therapy in neonates and maturation of the gut microbiome	Kuhn et al., 2022 [92]	“There are detectable benefits associated with breastfeeding in conjunction with early ART in maintaining a more bifidobacteria-rich microbiota profile in infants with HIV”.	Observational study
Dissection of the gut microbiota in mothers and children with chronic *Trichuris trichiura* infection in Pemba Island, Tanzania	Chen et al., 2021 [93]	“Changes in gut microbial composition and structure occur in *T. trichiura*-infected individuals compared with uninfected individuals”.	Comparative study
Interactions between fecal gut microbiome, enteric pathogens, and energy regulating hormones among acutely malnourished rural Gambian children	Nabwera et al., 2021 [94]	“Marasmus SAM is characterized by the collapse of a complex system with nested interactions and key associations between the gut microbiome, enteric pathogens, and energy regulating hormones”.	Cross-sectional observational sub-study of a quasi-experimental study
A signature of *Prevotella copri* and *Faecalibacterium prausnitzii* depletion, and a link with bacterial glutamate degradation in the Kenyan colorectal cancer patients	Obuya et al., 2022 [95]	“Microbiome and microbial metabolic profiles of CRC patients are different from those of healthy individuals. CRC microbiome dysbiosis, particularly *P. copri* and *F. prausnitzii* depletion and glutamate metabolic alterations, are evident in Kenyan CRC patients”.	Cross-sectional observational study
Gut microbiomes from Gambian infants reveal the development of a non-industrialized *Prevotella*-based trophic network	de Goffau et al., 2022 [96]	“Distinct bacterial trophic network clusters were identified, centred around either *P. stercorea* or *F. prausnitzii* and were found to develop steadily with age, whereas *P. copri*, independently of other species, rapidly became dominant after weaning”.	Sub-study of a clinical trial
Rectal microbiota diversity in Kenyan MSM is inversely associated with frequency of receptive anal sex, independent of HIV status	Gebrebrhan et al., 2021 [97]	“In addition to decreased alpha diversity in men with HIV infection, the authors also established a dose-dependent relationship between receptive anal sex with multiple partner types, controlling for HIV, and reduced rectal microbiome alpha diversity”.	Cross-sectional study
Stunted childhood growth is associated with decompartmentalization of the gastrointestinal tract and overgrowth of oropharyngeal taxa	Vonaesch et al., 2018 [98]	“Stunting is associated with a microbiome “decompartmentalisation” of the gastrointestinal tract characterized by an increased presence of oropharyngeal bacteria from the stomach to the colon”.	Transversal study
Gut microbiome profiling of a rural and urban South African cohort reveals biomarkers of a population in lifestyle transition	Oduaran et al., 2020 [17]	“The overall gut microbiome of the cohorts is reflective of their ongoing epidemiological transition. Geographical location was more important for sample clustering than lean/obese status and observed a relatively higher abundance of the *Melainabacteria*, *Vampirovibrio*, a predatory bacterium, in Bushbuckridge”.	Pilot study
Stool microbiota composition is associated with the prospective risk of *Plasmodium falciparum* infection	Yooseph et al., 2015 [99]	“The preliminary finding of an association between gut microbiota composition and *P. falciparum* infection risk suggests that strategic modulation of gut microbiota composition could decrease *P. falciparum* infection risk in malaria-endemic areas”.	Prospective cohort study
Longitudinal and comparative analysis of gut microbiota of Tunisian newborns according to delivery mode	Hanachi et al., 2022 [100]	“Both elective cesarean section (ECS) and vaginally delivered (VD) showed a profile dominated by *Proteobacteria*, *Actinobacteria*, and *Firmicutes*. However, ECS showed an underrepresentation of *Bacteroides* and an enrichment of opportunistic pathogenic species of the ESKAPE group, starting from the second week”.	Longitudinal study
Microbial gut evaluation in an Angolan paediatric population with sickle cell disease	Delgadinho et al., 2022 [101]	“Children with sickle cell disease (SCD) have a higher number of the phylum *Actinobacteria*. *Clostridium cluster XI* bacteria was more prevalent in the SCD children, whereas the siblings had a higher abundance of *Blautia*, *Aestuariispira*, *Campylobacter*, *Helicobacter*, *Polaribacter* and *Anaerorhabdus*”.	Cross-sectional study
How hydroxyurea alters the gut microbiome: A longitudinal study involving Angolan children with sickle cell anaemia	Delgadinho et al., 2022 [102]	“Hydroxyurea (HU) can influence the diversity and shape of the gut microbiome. However, it is not yet clear if the higher abundance of beneficial bacteria is a direct or indirect effect of HU treatment, being a consequence rather than a cause”.	Longitudinal study
Anaemia and iron status are predictors of gut microbiome composition and metabolites in infants and children in rural Kenya	Paganini et al., 2016 [103]	“In infancy, higher haemoglobin (Hb) and better iron status predict higher amounts of pathogenic *E. coli.* Among older children, higher Hb and better iron status predict lower amounts of pathogenic *E. coli*. Anaemic infants and children do not have greater gut inflammation but have lower faecal butyrate concentrations. Thus, relationships between anaemia, iron status and pathogenic gut microbiota differ by age in rural Kenya”.	Cross-sectional study
Iron fortification adversely affects the gut microbiome, increases pathogen abundance and induces intestinal inflammation in Kenyan infants	Jaeggi et al., 2015 [104]	“Provision of iron-containing micronutrient powder (MNPs) to weaning infants adversely affects the gut microbiome, increasing pathogen abundance and causing intestinal inflammation”.	Clinical trial
Atopic dermatitis and food sensitization in South African toddlers: Role of fibre and gut microbiota	Mahdavinia et al., 2017 [105]	“This pilot study found no major differences in the composition of the gut microbiota of 12- to 36-month-old children with and without AD”.	Pilot study
The gut microbiome but not the resistome is associated with urogenital schistosomiasis in preschool-aged children	Osakunor et al., 2020 [106]	“The authors identified differences in the gut microbiome between schistosome infected and uninfected children, showing largely an increase in abundance of specific bacteria”.	Cross-sectional study
The effect of oral iron supplementation on the gut microbiota, gut inflammation, and iron status in iron-depleted South African school-age children with virally suppressed HIV and without HIV	Goosen et al., 2022 [107]	“Oral iron supplementation can significantly improve haemoglobin and iron status without increasing pathogenic gut microbial taxa or gut inflammation in iron-depleted virally suppressed HIV+ and HIV−ve school-age children”.	Before-after intervention study with case–control comparisons
Impact of the post-transplant period and lifestyle diseases on human gut microbiota in kidney graft recipients	Souai et al., 2020 [108]	“The study shows specific kidney transplant-related effects of the fecal microbiome on graft stability and patient’s health status when compared to healthy subjects. The overall microbial community structure of the kidney transplanted group was clearly different from control subjects”.	Observational study
Mucosa-associated cultivable aerobic gut bacterial microbiota among colorectal cancer patients attending at the referral hospitals of Amhara regional state, Ethiopia	Siraj et al., 2021 [109]	“A relative abundance and distributions of cultivable aerobic bacterial microbiota of malignant tissues were significantly different from its adjacent normal tissue biopsies. Families of *Enterobacteriaceae* and *Enterococcaceae* were the most frequently recovered bacterial family from malignant tissues”.	Observational study
Iron in micronutrient powder promotes an unfavourable gut microbiota in Kenyan infants	Tang et al., 2017 [110]	“MNP fortification over three months in non- or mildly anaemic Kenyan infants can potentially alter the gut microbiome. Addition of iron to the MNP may adversely affect the colonization of potential beneficial microbes and attenuate the decrease of potential pathogens”.	Clinical trial
Impact of a nomadic pastoral lifestyle on the gut microbiome in the Fulani living in Nigeria	Afolayan et al., 2019 [111]	“Large differences in the taxonomic and functional composition in the intestinal microbiota between the Fulani and the urban Jarawa population were observed”.	Comparative study
Impact of geographical location on the gut microbiota profile in Egyptian children with type 1 diabetes mellitus: A pilot study	Elsherbiny et al., 2022 [112]	“The diabetic groups irrespective of the geographical location showed significantly lower alpha diversity, mean *Firmicutes*/*Bacteroidetes* (F/B) ratio, and reduced proportions of *Prevotella* and *Ruminococcus*. There were also significantly enriched representations of *Actinobacteria*, *Bacteroidetes*, and *Proteobacteria* and genera *Lactobacilli*, *Bacteroides*, and *Faecalibacterium* pointing to the greater driving power of the disease”.	Case-control study
Gut microbiota in forty cases of Egyptian relapsing remitting multiple sclerosis	Elgendy et al., 2021 [113]	“Changes in gut microbiota are associated with exacerbation of multiple sclerosis (MS) disease”.	Prospective study
Vitamin D and phenylbutyrate supplementation does not modulate gut derived immune activation in HIV-1	Missailidis et al., 2019 [114]	“Nutritional supplementation with vitamin D + phenylbutyrate did not modulate gut-derived inflammatory markers or microbial composition in treatment-naïve HIV-1 individuals with active viral replication”.	Clinical trial
Immunoglobulin recognition of fecal bacteria in stunted and non-stunted children: Findings from the Afribiota study	Huus et al., 2020 [115]	“Stunted children have a greater proportion of IgA-recognized fecal bacteria. Moreover, there is identification of two putative pathobionts, *Haemophilus* and *Campylobacter*, that are broadly targeted by intestinal IgA”.	Observational study
Exploring the relationship between the gut microbiome and mental health outcomes in a posttraumatic stress disorder cohort relative to trauma-exposed controls	Malan-Muller et al., 2022 [116]	“*Mitsuokella*, *Odoribacter*, *Catenibacterium* and *Olsenella*, possess a moderate ability to discriminate between PTSD and TEC participants. The summed relative abundance of these genera was higher in individuals with PTSD compared to TEC and was positively correlated with PTSD severity and childhood trauma severity”.	Case-control study
The Effect of ss-glucan prebiotic on kidney function, uremic toxins, and gut microbiome in stage 3 to 5 chronic kidney disease (CKD) pre-dialysis participants: A randomised controlled trial	Ebrahim et al., 2022 [117]	“The supplementation of ß-glucan fibre resulted in favourable change in the composition of the gut microbiome. The authors therefore reject the hypothesis that kidney function would change with the prebiotic and accept the hypothesis that the gut microbiome improved with the prebiotic intervention”.	Clinical trial
Modifying gut integrity and microbiome in children with severe acute malnutrition using legume-based feeds (MIMBLE): A pilot trial	Calder et al., 2021 [118]	“Cowpea [Cp]-enriched feeds [F] [CpF]) has a positive effect on the fecal microbiota, particularly between days 1 and 7”.	Pilot study
Gut microbiome function and composition in infants from rural Kenya and association with human milk oligosaccharides	Derrien et al., 2023 [119]	“Gut microbiome of partially breastfed Kenyan infants over the age of six months is enriched in bacteria from the *Bifidobacterium* community, including *B. infantis*, and the high prevalence of a specific HM group may indicate a specific HMO-gut microbiome association”.	Observational study
Multi-nutrient fortified dairy-based drink reduces anaemia without observed adverse effects on gut microbiota in anaemic malnourished Nigerian toddlers: A randomised dose–response study	Owolabi et al., 2021 [120]	“Daily consumption of 200–600 mL of iron-fortified multi-nutrient fortified dairy-based drink reduces anaemia without stimulating potential pathogenic gut bacteria in Nigerian toddlers”.	Clinical trial
Antenatal gut microbiome profiles and effect on pregnancy outcome in HIV infected and HIV uninfected women in a resource limited setting	Chandiwana et al., 2023 [121]	“Species richness and taxonomy composition of the gut microbiota is altered in HIV-infected pregnant women, possibly reflecting intestinal dysbiosis. Some of the taxa were also associated with low infant birth weight”.	Cross-sectional sub-study
Evolution of the gut microbiome in HIV-exposed uninfected and unexposed infants during the first year of life	Jackson et al., 2022 [122]	“Gut microbiomes of HIV-exposed uninfected (HEU) and HIV-unexposed uninfected (HUU) are heavily influenced by the maternal gut microbiome. The microbiotas of HEU and HUU converged over time, mirroring the decrease in the excess of infectious morbidity and mortality in HEU”.	Longitudinal cohort
Breastmilk, stool, and meconium: bacterial communities in South Africa	Wallenborn et al., 2022 [123]	“Despite the importance of breastmilk in seeding the infant gut microbiome, the authors found evidence of distinct bacterial communities between breastmilk and stool samples from South African mother-infant dyads”.	Pilot study
Robust variation in infant gut microbiome assembly across a spectrum of lifestyles	Olm et al., 2022 [124]	“Infants from all lifestyles begin life with similar bifidobacteria-dominated gut microbiota compositions, but subtle differences detected in early life compound over time”.	Observational study
Intestinal protozoan infections shape fecal bacterial microbiota in children from Guinea-Bissau	von Huth et al., 2021 [125]	“Infections with multiple parasite species induces more pronounced changes in the bacterial composition, and certain parasite species significantly changes the diversity of bacteria”.	Prospective no-intervention two-cohort study
Evolution of the gut microbiome following acute HIV-1 infection	Rocafort et al., 2019 [126]	“Recent HIV-1 infection is associated with increased fecal shedding of eukaryotic viruses, transient loss of bacterial taxonomic richness, and long-term reductions in microbial gene richness. An HIV-1-associated microbiome signature only becomes evident in chronically HIV-1-infected subjects”.	Sub-study of a prospective observational cohort study
Growth velocity in children with environmental enteric dysfunction is associated with specific bacterial and viral taxa of the gastrointestinal tract in Malawian children	Desai et al., 2020 [127]	“The data of the study demonstrate associations between specific bacterial and viral taxa and growth velocity, though the authors cannot yet prove a causal relationship exists between these measures”.	Longitudinal cohort study
Gut microbiota signature of pathogen-dependent dysbiosis in viral gastroenteritis	Mizutani et al., 2021 [128]	“This study revealed specific trends in the gut microbiota signature associated with diarrhoea and that pathogen-dependent dysbiosis occurred in viral gastroenteritis. It also revealed that several bacterial taxa with potential pathogenesis, such as *Escherichia-Shigella* and *Klebsiella*, are part of healthy commensal microbiota in Ghanaian individuals”.	Comparative study
Prebiotic galacto-oligosaccharides mitigate the adverse effects of iron fortification on the gut microbiome: A randomised controlled study in Kenyan infants	Paganini et al., 2017 [129]	“A MNP containing a 5 mg daily dose of highly bioavailable iron is likely safer in that it induces less adverse changes in the gut microbiome and no increase in faecal calprotectin compared with a 12.5 mg iron dose”.	Clinical trial
*Blastocystis* colonization is associated with increased diversity and altered gut bacterial communities in healthy Malian children	Kodio et al., 2019 [130]	“*Blastocystis* colonization was significantly associated with a higher diversity and richness of the gut bacterial communities in healthy children. Also, *Blastocystis* colonization was associated with a higher proportion of “beneficial bacteria” (*Firmicutes* and *Bacteroides*) than “probable pathogenic bacteria” (*Proteobacteria*) in the human gut”.	Cross-sectional study
Association between clinical and environmental factors and the gut microbiota profiles in young South African children	Nel Van Zyl et al., 2021 [131]	“*Prevotella* was the most common genus identified in the participants, and after infancy, the gut bacteria were dominated by *Firmicutes* and *Bacteroidetes*. In this setting, children exposed to antibiotics and indoor cooking fires were at the most risk for dysbiosis, showing significant losses in gut bacterial diversity”.	Sub-study of a clinical trial
Fecal microbiome composition in healthy adults in Ghana	Parbie et al., 2021 [132]	“The fecal microbiome of the Ghanaian adults was dominated by *Firmicutes* (*Faecalibacterium*, *Subdoligranulum*, and *Ruminococcaceae* UCG-014), *Proteobacteria* (*Escherichia*-*Shigella* and *Klebsiella*), and *Bacteroidetes* (*Prevotella* 9 and *Bacteroides*), consistent with previous observations in African cohorts”.	Cross-sectional study
Dysbiotic fecal microbiome in HIV-1 infected individuals in Ghana	Parbie et al., 2021 [133]	“Analysis revealed significant difference in fecal microbiome diversity and compositions between HIV-1 infected and uninfected individuals. Particularly, the study revealed the characteristics of dysbiotic fecal microbiome in HIV-1 infected adults in Ghana”.	Cross-sectional case-control study
Mass azithromycin distribution and community microbiome: A cluster-randomised trial	Doan et al., 2018 [134]	“Two mass azithromycin administrations, 6 months apart, in preschool children led to long-term alterations of the gut microbiome structure and community diversity. Long-term microbial alterations in the community did not imply disease but were associated with an improvement in childhood mortality”.	Clinical trial
Gut microbial diversity in antibiotic-naive children after systemic antibiotic exposure: A randomised controlled trial	Doan et al., 2017 [135]	“The paediatric gut microbiome is sensitive to antibiotic exposure. A single dose of antibiotic can result in a significant reduction in bacterial diversity”.	Clinical trial
High-throughput sequencing of pooled samples to determine community-level microbiome diversity	Ray et al., 2019 [136]	“Pooling microbiome samples before DNA amplification to estimate community level diversity is a viable and valuable measure to consider in population-level association research studies”.	Observational study
Environmental enteric dysfunction and the fecal microbiota in Malawian children	Ordiz et al., 2017 [137]	“Environmental enteric dysfunction (EED) is not associated with a profound fecal dysbiosis, but six genera were identified as having significantly different abundances in EED.”	Clinical trial
The effect of legume supplementation on the gut microbiota in rural Malawian infants aged 6 to 12 months	Ordiz et al., 2020 [138]	“Extensive 16S sequencing of fecal samples from a trial of cowpea supplementation in rural African infants did not reveal a microbiota signature for stunting. The association of the *Veillonella* genus in the feces with health benefits in several populations warrants further investigation”.	Clinical trial
Higher fibre complementary food alters fecal microbiota composition and normalises stool form in Malawian children: A randomised trial	Lungu et al., 2021 [139]	“In young Malawian children, feeding a blend of soybean, soy hulls, and maize reduced diarrhoea-type stools and increased the abundance of *Akkermansia muciniphila*, a bacterial species involved in maintaining intestinal health”.	Clinical trial
Co-trimoxazole reduces systemic inflammation in HIV infection by altering the gut microbiome and immune activation	Bourke et al., 2019 [140]	“Co-trimoxazole reduces systemic and intestinal inflammation both indirectly via antibiotic effects on the microbiome, and directly by blunting immune and epithelial cell activation”.	Experimental and longitudinal study
Effect of radiotherapy on the gut microbiome in pediatric cancer patients: A pilot study	Sahly et al., 2019 [141]	“A correlation between microbial composition and response to treatment was reported, in which the responders had generally a lower microbial diversity compared to non-responders. In addition, nucleotide changes and deletions in the tested 16S rRNA sequences post radiotherapy were detected”.	Pilot study
Environmental exposures and child and maternal gut microbiota in rural Malawi	Kortekangas et al., 2020 [142]	“In this study in a rural Malawian setting, environmental exposures were associated with several subtle aspects of microbiota composition, but authors did not find consistent associations with microbiota maturity or diversity”.	Sub-study of a clinical trial
An alternative oat-containing, ready-to-use, therapeutic food does not alter intestinal permeability or the 16S ribosomal RNA fecal microbiome configuration among children with severe malnutrition in Sierra Leone: A randomised controlled trial	Hendrixson et al., 2023 [143]	“Despite remarkably different compositions of oat ready-to-use therapeutic food (o-RUTF) and standard RUTF (s-RUTF), no differences were identified in lactulose permeability or the fecal 16S rRNA configuration among children with SAM receiving these foods”.	Clinical trial
Effects of diet on the childhood gut microbiome and its implications for atopic dermatitis	Mahdavinia et al., 2019 [144]	“The authors have identified 2 species of bacteria that seem to be linked to AD and diet and can explain some of the unsolved chain of events in the pathogenesis of this condition”.	Cross-sectional study
Co-trimoxazole prophylaxis increases resistance gene prevalence and α-diversity but decreases β-diversity in the gut microbiome of human immunodeficiency virus–exposed, uninfected infants	D’Souza et al., 2020 [145]	“Co-trimoxazole prophylaxis in HEU infants decreased gut microbiome β-diversity and increased antibiotic resistance gene α-diversity and prevalence”.	Experimental study
Alterations of gut microbiota in type 2 diabetes individuals and the confounding effect of antidiabetic agents	Almugadam et al., 2020 [146]	“The study revealed a significantly lowered abundance of *Faecalibacterium*, *Fusobacterium*, *Dialister*, and *Elusimicrobium* in the nontherapeutic T2DM subgroup. Correlation analysis showed a substantial decline in gut microbiota richness and diversity with the duration of illness. Antidiabetic agents restored to some extent the richness and diversity of gut microbiota and improved the abundance of many beneficial bacteria”.	Case-control study
Impact of azithromycin mass drug administration on the antibiotic-resistant gut microbiome in children: A randomised, controlled trial	Pickering et al., 2022 [147]	“This study found significant changes in the antimicrobial resistance profile and gut microbiome after four biannual rounds of azithromycin. Abundance of enteropathogen *E. albertii* was increased after treatment, as well as several opportunistic *Acinetobacter* pathogens”.	Clinical trial
Variation in rural African gut microbiota is strongly correlated with colonization by *Entamoeba* and subsistence	Morton et al., 2015 [148]	“This study suggests an important role for eukaryotic gut inhabitants and the potential for feedback between helminths, protozoa, microbes, and the host immune response, one that has been largely overlooked in studies of the microbiome”.	Observational study
Gut bacteria missing in severe acute malnutrition, can we identify potential probiotics by culturomics?	Alou et al., 2017 [149]	“The authors found a globally decreased diversity, a decrease in the hitherto unknown diversity, a depletion in oxygen-sensitive prokaryotes including *Methanobrevibacter smithii* and an enrichment in potentially pathogenic *Proteobacteria*, *Fusobacteria* and *Streptococcus gallolyticus*. A complex of 12 species identified only in healthy children using culturomics and metagenomics were identified as probiotics candidates”.	Case-control
Gut microbiota in children hospitalized with oedematous and non-oedematous severe acute malnutrition in Uganda	Kristensen et al., 2016 [150]	“The authors found that non-oedematous SAM children have lower GM diversity compared to oedematous SAM children, but no clear compositional differences were found between the two groups of children”.	Observational study
Growth and morbidity of Gambian infants are influenced by maternal milk oligosaccharides and infant gut microbiota	Davis et al., 2017 [151]	“While bifidobacteria were the dominant genus in the infant gut overall, *Dialister* and *Prevotella* were negatively correlated with morbidity, and *Bacteroides* was increased in infants with abnormal calprotectin”.	Experimental study
The seasonal changes of the gut microbiome of the population living in traditional lifestyles are represented by characteristic species-level and functional-level SNP enrichment patterns	Zhu et al., 2021 [152]	“Eight prevalent species have significant single nucleotide polymorphism (SNP) enrichments with the increasing number of SNP, among which only *Eubacterium biforme*, *Eubacterium hallii* and *Ruminococcus obeum* have relatively high species abundances. Many genes in the microbiomes also presented characteristic SNP distributions between the wet season and the dry season”.	Observational study
Breast milk and gut microbiota in African mothers and infants from an area of high HIV prevalence	González et al., 2013 [153]	“Both breast milk and faecal microbiota composition varied with lactation period, which might be related to changes in the type of feeding over time and/or in the milk’s biochemical characteristics”.	Cross-sectional study
Gut microbiome of Moroccan colorectal cancer patients	Allali et al., 2018 [154]	“CRC stools were markedly different from controls, showing an overrepresentation of 33 genera”.	Observational study
Gastrointestinal, vaginal, nasopharyngeal, and breast milk microbiota profiles and breast milk metabolomic changes in Gambian infants over the first two months of lactation: A prospective cohort study	Karampatsas et al., 2022 [155]	“This study’s results indicate that infant gut microbiota are unique bacterial communities, distinct from maternal gut and breast milk, respectively”.	Prospective cohort study
Response of the human gut and saliva microbiome to urbanization in Cameroon	Lokmer et al., 2020 [156]	“Urbanization was associated with minor shifts in diversity of the gut and saliva microbiome, but also with changes in the gut microbiome composition that were reminiscent of those associated with industrialization”.	Observational study
Gut microbiome of coexisting BaAka pygmies and Bantu reflects gradients of traditional subsistence patterns	Gomez et al., 2016 [157]	“Although the microbiome of both groups is compositionally similar, hunter-gatherers harbor increased abundance of *Prevotellaceae*, *Treponema*, and *Clostridiaceae*, while the Bantu gut microbiome is dominated by *Firmicutes*”.	Observational study
Gut microbiota impact on Angolan children with sickle cell disease	Brito et al., 2022 [158]	“The SCD and control samples exhibited some notable differences in microbiota relative abundance, at different levels of classification”.	Observation-al study
The effect of dietary resistant starch type 2 on the microbiota and markers of gut inflammation in rural Malawi children	Ordiz et al., 2015 [159]	“Resistant starch (RS) does not confer physiologically meaningful changes on gut biology, or gut microbial content, after short course treatment. The study data do not preclude the potential use of other prebiotics but provide no data in support of using RS to improve gut health in the studied sub-Saharan childhood population”.	Pilot study
Gut microbiota in HIV–pneumonia patients is related to peripheral CD4 counts, lung microbiota, and in vitro macrophage dysfunction	Shenoy et al., 2019 [160]	“This study is the first to reveal a relationship between gut microbiota composition and CD4 status. Moreover, it provides the first evidence that the products of the gut microbiome of HIV-infected patients with low CD4 counts alter monocyte effector phenotypes, reducing their capacity for repair and promoting a program of pro-inflammatory activity”.	Cross-sectional study
Differences in fecal gut microbiota, short-chain fatty acids and bile acids link colorectal cancer risk to dietary changes associated with urbanization among Zimbabweans	Katsidzira et al., 2019 [161]	“The gut microbiota composition and activity among rural and urban Zimbabweans retain significant homogeneity (possibly due to retention of dietary fibre), but urban residents have subtle changes, which may indicate a higher CRC risk”.	Exploratory study
Changes in the human gut microbiota associated with colonization by *Blastocystis* sp. and *Entamoeba* spp. in non-industrialized populations	Even et al., 2021 [162]	“The authors strongly suggest that *Blastocystis* and *Entamoeba* likely act through different mechanisms to interact with the gut bacterial microbiota”.	Observational study
Gut microbiome alteration in MORDOR I: A community-randomised trial of mass azithromycin distribution	Doan et al., 2019 [163]	“The same mass drug administration (MDA) that caused a reduction in childhood mortality also reduced certain gut bacteria, including known pathogens. Specifically, azithromycin distribution altered microbial metabolic function and decreased organisms known to cause human disease”.	Clinical trial
Gut microbiota alteration is characterized by a *Proteobacteria* and *Fusobacteria* bloom in Kwashiorkor and a Bacteroidetes paucity in Marasmus	Pham et al., 2019 [164]	“The kwashiorkor gut microbiota was characterized by an increased proportion of *Proteobacteria*, but there was a decreased proportion of *Bacteroidetes* in marasmus. *Fusobacterium* was more frequently cultured from kwashiorkor. All detected potential pathogenic species were enriched in the kwashiorkor gut microbiota”.	Observational study
Patients infected with *Mycobacterium africanum* versus *Mycobacterium tuberculosis* possess distinct intestinal microbiota	Namasivayam et al., 2020 [165]	“*Mycobacterium africanum* (MAF) participants have distinct microbiomes compared with *Mycobacterium tuberculosis* (MTB) patients, displaying decreased diversity and increases in *Enterobacteriaceae* with respect to healthy participants not observed in the latter patient group”.	Longitudinal study
MALDI-TOF identification of the human gut microbiome in people with and without diarrhoea in Senegal	Samb-Ba et al., 2014 [166]	“In individuals with diarrhoea, major commensal bacterial species such as *E. coli* were significantly decreased (85% versus 64%), as were several *Enterococcus* spp. (*E. faecium* and *E. casseliflavus*) and anaerobes, such as *Bacteroides* spp. (*B. uniformis* and *B. vulgatus*) and *Clostridium* spp. (*C. bifermentans*, *C. orbiscindens*, *C. perfringens*, and *C. symbosium*). Conversely, several *Bacillus* spp. (*B. licheniformis*, *B. mojavensis*, and *B. pumilus*) were significantly more frequent among patients with diarrhoea”.	Comparative study
Associations of fecal microbial profiles with breast cancer and non-malignant breast disease in the Ghana breast health study	Byrd et al., 2021 [167]	“Alpha diversity, overall microbiota composition, and taxa with hypothesized estrogen-conjugation and immune-related functions may be associated with breast diseases”.	Case-control study
The oral microbiome and breast cancer and non-malignant breast disease, and its relationship with the fecal microbiome in the Ghana breast health study	Wu et al., 2022 [168]	“Some periodontal pathogens are inversely associated with breast cancer and nonmalignant breast disease. Additionally, the oral and fecal microbiome appear to be more correlated among women with breast cancer or nonmalignant breast disease compared to controls”.	Case-control study
Feeding-related gut microbial composition associates with peripheral T-cell activation and mucosal gene expression in African infants	Wood et al., 2018 [169]	“Non-exclusive breastfeeding alters the gut microbiota, increasing T-cell activation and, potentially, mucosal recruitment of HIV target cells”.	Prospective, longitudinal study
Dynamics of the gut microbiome in *Shigella*-infected children during the first two years of life.	Ndungo et al., 2022 [170]	“*Shigella* infection did not profoundly impact overall species diversity but led to the expansion of species known to improve gastrointestinal health and drive the microbiota back to homeostasis”.	Longitudinal study
A prebiotic-enhanced lipid-based nutrient supplement (LNSp) increases *Bifidobacterium* relative abundance and enhances short-chain fatty acid production in simulated colonic microbiota from undernourished infants	Toe et al., 2020 [171]	“Provision of prebiotic enhanced LNS to undernourished children could be a possible strategy to steer the microbiota toward a more beneficial composition and metabolic activity”.	Observational study
HIV-exposure, early life feeding practices and delivery mode impacts on faecal bacterial profiles in a South African birth cohort	Claassen-Weitz et al., 2018 [172]	“Major determinants of infant meconium bacterial profiles were mode of feeding and maternal body mass index (BMI). HIV-exposure, on the other hand, was an important contributor to the composition of infant faecal bacterial profiles at 4–12 weeks of life, with HIV-exposed infants having higher bacterial diversity and reduced proportions of bifidobacteria”.	Observational pilot study
The fecal microbiome and rotavirus vaccine immunogenicity in rural Zimbabwean infants	Robertson et al., 2021 [173]	“The authors found no clear stool microbiome signature associated with oral rotavirus vaccine (RVV) immunogenicity in rural Zimbabwean infants”.	Observational study
The impact of storage conditions on human stool 16S rRNA microbiome composition and diversity	Carruthers et al., 2019 [174]	“Stool storage preservation method significantly influenced the bacterial profiles obtained, however, all samples remained identifiable to their child of origin. Stool stored at ambient temperature for up to 32 h did not significantly influence diversity and had minimal changes upon microbiota composition, which remained relatively stable across time-to-freezing regardless of preservation method used”.	Observational study
Short- and long-read metagenomics of urban and rural South African gut microbiomes reveal a transitional composition and undescribed taxa	Tamburini et al., 2022 [175]	“Gut microbiome of South Africans does not conform to a simple “western-nonwestern” axis and contains undescribed microbial diversity”.	Observational study
Associations between gut microbiota and intestinal inflammation, permeability, and damage in young Malawian children	Kortekangas et al., 2022 [176]	“The study findings support the hypothesis of an association between the gut microbiota composition and EED assessed by the biomarkers calprotectin, alpha-1 antitrypsin and REG1B in rural Malawian children”.	Observational study
Gut microbiota imbalances in Tunisian participants with type 1 and type 2 diabetes mellitus	Fassatoui et al., 2019 [177]	“There was a reduction in the amount of *A. muciniphila*, *Firmicutes*, and *F. prausnitzii* in Tunisian participants with diabetes. The abundance of *A. muciniphila* was also affected by glycemic level”.	Comparative study
Effect of commonly used pediatric antibiotics on gut microbial diversity in preschool children in Burkina Faso: A randomised clinical trial	Oldenburg et al., 2018 [178]	“A short course of azithromycin led to a significant decrease in bacterial diversity of the intestinal microbiome in preschool children. Although the clinical implications of a single course of antibiotics are unclear, the intestinal microbiome in young children is sensitive to antibiotics”.	Clinical trial
Associations of HIV and iron status with gut microbiota composition, gut inflammation and gut integrity in South African school-age children: A two-way factorial case–control study	Goosen et al., 2023 [179]	“In 8-to 13-year-old virally suppressed HIV+ and HIV− children with or without iron deficiency (ID), ID was associated with increased gut inflammation and changes in the relative abundance of specific microbiota. In HIV+ children, ID had a cumulative effect that further shifted the gut microbiota to an unfavourable composition”.	Two-way factorial case-control study

## Data Availability

Data are all available in the manuscript.

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
