# Peer review of "A Scoping Review Evaluating the Current State of Gut Microbiota Research in Africa"

_microorganisms, 2023, doi:10.3390/microorganisms11082118_

Round 1
Reviewer 1 Report
GENERAL COMMENTS
Although the writing is quite good, there are some errors. Thus, the article should be reviewed by a native English speaker.
TITLE
OK.
ABSTRACT
OK.
INTRODUCTION
Please check grammar of the first sentence: “The gut microbiota, a collection of microorganisms that inhabit the human gastrointestinal system search in recent years”.
Some other typos are detected throughout the text (such as the incorrect use of capital letters, etc.). Please carefully review the entire manuscript.
METHODS
Criteria for exclusion: Studies that did not involve human participants or included both humans and animals were excluded from the review. Why were these last studies excluded (at least data on humans could have been used)?
Was any language restriction for a study to be included? This information should be provided in the Methods section.
RESULTS
OK.
DISCUSSION
OK, comprehensive review.
Please check this probable text mistake: “Top of Form” in page 33.
A section summarising the strengths and, especially, the potential limitations of the present study is lacking.
REFERENCES
OK.
TABLES
If there are space limitations, consider deleting Table 2 (General/key findings of the included studies), as it is very extensive (although includes quite relevant information).
FIGURES
If the number of figures is considered (by the Editor) to be excessive, consider deleting Figure 7 (Distribution of sample types utilized for gut microbiota profiling), as its information can be easily included in the text.
Figure 4. Some colours are hard to distinguish (for example some shades of green); please consider adding or changing some of them.
Although the writing is quite good, there are some errors. Thus, the article should be reviewed by a native English speaker.
Author Response
Please see attached letter.

Reviewer 2 Report
The paper is interesting since it gives information regarding the state of the art of Microbiome research in Africa. My only concern is table 2, whcih is too long and impossible to access. I suggest authors shoud condensate the key message at least as far as column 3 is concerned (General/ key findings of the studies) to a maximum of 20 words or so.
Author Response
Please see attached letter.
